# STATIONARY DEEP REINFORCEMENT LEARNING WITH QUANTUM K-SPIN HAMILTONIAN EQUATION

## ABSTRACT

Instability is a major issue of deep reinforcement learning (DRL) algorithms —
high variance of cumulative rewards over multiple runs. The instability is mainly
caused by the existence of *many local minimas* and worsened by the *multiple fixed
points* issue of Bellman's optimality equation. As a fix, we propose a quantum
K-spin Hamiltonian regularization term (called *H-term*) to help a policy network
converge to a high-quality local minima. First, we take a quantum perspective by
modeling a policy as a *K-spin Ising model* and employ a Hamiltonian equation
to measure the *energy* of a policy. Then, we derive a novel Hamiltonian policy
gradient theorem and design a generic actor-critic algorithm that utilizes the H-
term to regularize the policy network. Finally, the proposed method significantly
reduces the variance of cumulative rewards by $65.2\% \sim 85.6\%$ on six MuJoCo
tasks; achieves an approximation ratio $\leq 1.05$ over $90\%$ test cases and reduces its
variance by $60.16\% \sim 94.52\%$ on two combinatorial optimization tasks and two
non-convex optimization tasks, compared with those of existing algorithms over
20 runs, respectively.

## 1 INTRODUCTION

Instability is a major issue of deep reinforcement learning (DRL) [44] algorithms — agents trained
with different random seeds may have dramatically different performance. Existing works [1, 8, 16,
28, 31, 53] empirically reported a high variance over multiple runs. Hence, in practice it requires
to train tens of agents and pick the best one. Such a high variance largely contributes to the RL
community's dispute of reliability and reproducibility [17, 18], limiting the wider adoption in real-
world tasks. The instability issue is mainly caused by the existence of *many local minimas*[1] and
worsened by the *multiple fixed points* issue of Bellman's optimality equation [5, 21, 26, 39]. In Fig. 1,
we adapt dynamic programming examples [5, 39] into reinforcement learning settings, while detailed
descriptions are given in Appx. A.

- **Shortest path problem (deterministic)** in Fig. 1(a): two policies, 1) transiting back to state 1; 2)
  driving to terminal state 0.
- **Blackmailer's problem (stochastic)** in Fig. 1(b): two policies, 1) demanding $a \rightarrow 0$ to keep the
  victim at state 1; 2) demanding $a = 1$ that drives the victim to terminate state 0.
- **Optimal stopping problem (terminating policies)** in Fig. 1(c): two polices, 1) continuing inside
  the sphere of radius $(1 - \alpha)c$ and stopping outside; 2) jumping to point 0 at any point in region $C$.

The instability problem has been partially addressed, such as ensemble methods [2, 10], regulariza-
tion approaches [11, 46], and baseline-correction approaches [41, 50]. In particular, Generalized
Advantage Estimation (GAE) [41] is a widely used one that significantly reduces the variance of the
advantage function. However, they did NOT fix the issue of local minimas and the multiple fixed
points issue of Bellman equation in Fig. 1. Existing methods randomly converge to different local
minimas. For practical usage, we often expect a DRL algorithm stably converges to a certain policy
independent of initialization and noises.

As a fix, we propose a quantum K-spin Hamiltonian regularization term (*H-term*) to help a policy
network converge to a high-quality local minima. We take a novel quantum perspective by modeling

---

[1]Without explicit clarifications, both "local minima" and "fixed points" in this paper are referring to policies.

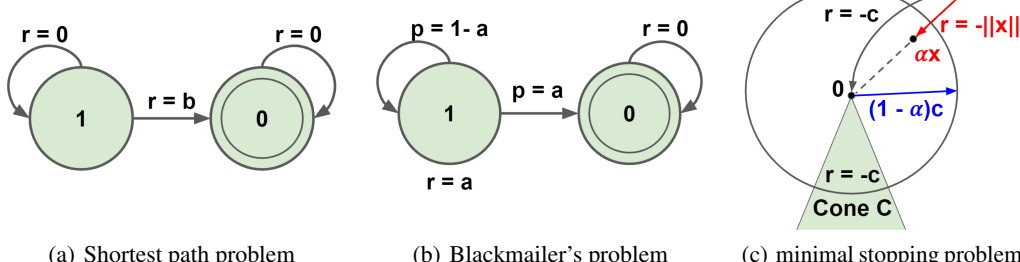

(a) Shortest path problem   (b) Blackmailer's problem   (c) minimal stopping problem

Figure 1: Examples with $\gamma = 1$. Examples with $\gamma < 1$ are given in Fig. 6 of Appx. A.

a policy as a *K-spin Ising model* [15, 30] and employ a Hamiltonian equation to measure the *energy* of a policy, namely an *H-term*. We hypothesize that *a stationary policy would have a low energy*.

In this paper, we propose a quantum K-spin Hamiltonian regularization term (called *H-term*) to help a policy network converge to a high-quality local minima. Our contributions can be summarized as follows: 1) we take a quantum perspective by modeling a policy as a *K-spin Ising model* and employ a Hamiltonian equation to measure the *energy* of a policy, which becomes an add-on term to DRL algorithms; 2) we derive a novel Hamiltonian policy gradient theorem and design a generic actor-critic algorithm that utilizes the H-term to regularize the policy/actor network; 3) we show that the proposed method significantly reduces the variance of cumulative rewards by $65.2\% \sim 85.6\%$ on six challenging MuJoCo tasks [47]; achieves an approximation ratio $\leq 1.05$ over $90\%$ test cases and reduces the variance of approximation ratio by $60.16\% \sim 94.52\%$ on two combinatorial optimization tasks (travelling salesman problem [31], graph maxcut [14]) and two non-convex optimization tasks (MIMO beamforming in 5G/6G [7], non-convex deep learning classifier [33]), compared with those of existing algorithms over 20 runs, respectively.

## 2 RELATED WORKS

The existence of many local minimas has been theoretically pointed out in robotic control tasks [16], combinatorial optimization tasks [25][36], and non-convex optimization tasks [3][52]. Existing solutions can be classified three approaches, ensemble method, regularizer, and basline-correction.

The ensemble method [2, 10] was proposed to reduce the variance by using multiple critic networks to approximate an accurate value function. However, this method will still encounter the multiple fixed points issue of Bellman's optimality equation. Regularization method [11, 46] was proposed to guide the updating process of a policy network. Adding a regularizer essentially helps find a local minima with preferred structure, which cannot help escape from local minimas.

Baseline-correction approaches [41, 50] was used to reduce the bias of monte carlo estimation. In particular, Generalized Advantage Estimation (GAE) [41] is a widely used one that significantly reduces the variance of the advantage function. However, the method is restricted by the accuracy of the baseline, which suffers from the local minimas issue as well.

However, they did NOT fix the issue of many local minimas and the multiple fixed points issue of Bellman equation in Fig. 1. In contrast, we propose a physically inspired DRL algorithm that stably converges to a certain policy independent of initialization and noises.

Different from our quantum K-spin perspective, several recent papers utilized the (classical) Hamiltonian equation to endow RL agents the capability of inductive biases. For example, [24, 48] used Hamiltonian mechanics to train an agent that learns and respects *conservation laws*; [51] applied a Hamiltonian Monte Carlo (HMC) simulator to approximate the posterior action probability; and [35] proposed an unbiased estimator for the stochastic Hamiltonian gradient methods for min-max optimization problems.

## 3 THE PROBLEM OF MANY LOCAL MINIMAS

First, we show the existence of many local minimas in many tasks. Then, we provide observational experiments to empirically verify the existence of multiple policies.

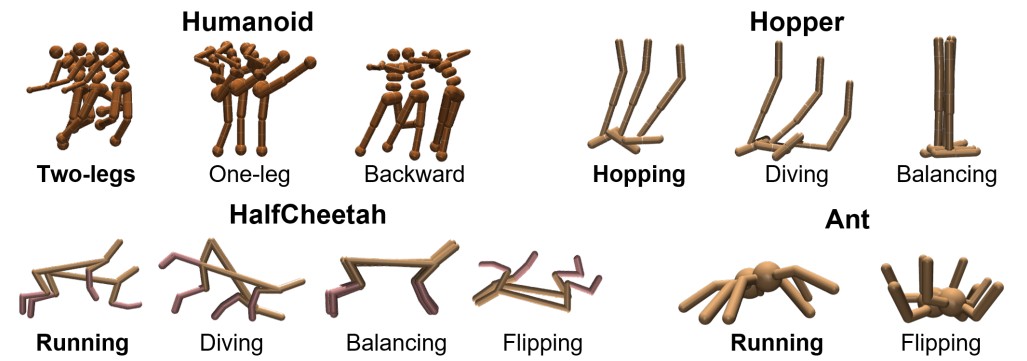

Figure 2: Different policies for MuJoCo tasks [47]. The bold ones are physically stationary policies.

### 3.1 Existence of Many Local Minimas

We point out that combinatorial optimization and non-convex optimization have many local minimas.

- MuJoCo tasks [47]: agents randomly converge to policies of different gaits, as shown in Fig. 2.
- Travelling salesman problem (TSP) [31]: a case of 8 cities has 2 local minimas (in Appx. K).
- Graph max-cut [14]: an example graph of 20 nodes has 390 local minimas (in Appx. K).
- MIMO beamforming [7]: a case of 2 users and 2 antennas has 3 local minimas (in Appx. L).
- Non-convex deep learning classifier [33]: an example problem has 25 local minimas [3].

### 3.2 Multiple Policies of Existing DRL Algorithms

We provide observational experiments on four challenging MuJoCo tasks [47], namely, Humanoid, Hopper, HalfCheetah, and Ant (details given in Appx. B.1), which are typical examples of the locomotion control of a robot. We render the obtained policies over multiple runs and then identify physically stationary ones. We observe various types of moving strategies, as shown in Fig. 2, which verifies that multiple policies are very common. For example, the Humanoid agent learns either jumping with a single leg or running with two legs, as shown in Fig. 2 (top-left); another interesting example is HalfCheetah, in which an agent can run normally or in a flipped manner, as shown in Fig. 2 (bottom-left). Among the obtained policies, one can easily identify the physically stationary polices that control the robot moving forward with a *stable gait* (defined as gait that does not lead to fall).

## 4 Modeling Policy as K-spin Ising Model

We take a novel quantum perspective by modeling a policy as a *K-spin Ising model* and employ a Hamiltonian equation to measure the *energy* of a policy, namely an *H-term*.

### 4.1 Motivation

Our modeling a policy as a quantum K-spin Ising model is inspired by the **simulated annealing** algorithms and the analogy in Table 1. Simulated annealing algorithms randomly transit to a neighbor solution with probability proportional to the energy gap between the current state and a new state. Here, a state can be modelled as a spin configuration, and a Hamiltonian equation is used to measure the energy of a spin system. Take the graph maxcut problem as an example, a spin is a configuration of nodes, while a transition (in terms of a state-action pair) is taken according to a policy. Using the modelling in Table 1, we learn a policy network that encodes the transition probability of a simulated annealing algorithm.

### 4.2 Quantifying Energy of a Policy via K-spin Hamiltonian Equation

The Hamiltonian equation for a quantum K-spin Ising model [15, 30] measures the energy of a particular configuration, which takes the following form

$$H = -\sum_{k=0}^{K-1} \sum_{j_0=1}^{N} \cdots \sum_{j_k=1}^{N} L_{j_0 \cdots j_k} \sigma_{j_0} \cdots \sigma_{j_k}, \tag{1}$$

Table 1: Modeling a policy as a quantum K-spin Ising model.

| | Policy in (3) | | Quantum K-spin Ising model [15, 30] in (1) | |
|---|---|---|---|---|
| State-action | $\mu_k \in \mathcal{S} \times \mathcal{A}, k = 0, ..., K$ | | Spins | $j_k \in \{1, \cdots, N\}, k = 0, ..., K$ |
| Policy | $\pi_{\mu_0} \times \pi_{\mu_1} \times \cdots \times \pi_{\mu_{K-1}} \in [0,1]^K$ | | Configuration | $\sigma_{j_0} \times \sigma_{j_1} \times \cdots \times \sigma_{j_{K-1}} \in [-1,+1]^K$ |
| Optimal policy | $\pi_{\mu_0}^* \times \pi_{\mu_1}^* \times \cdots \times \pi_{\mu_{K-1}}^* \in \{0,1\}^K$ | | Optimal configuration | $\sigma_{j_0}^* \times \sigma_{j_1}^* \times \cdots \times \sigma_{j_{K-1}}^* \in \{-1,+1\}^K$ |
| Discounted reward | $L_{\mu_0 \ldots \mu_{K-1}}$ | | Density function | $L_{j_0 \cdots j_{K-1}}$ |
| Energy of policy | $H(\pi_{\mu_0}, ..., \pi_{\mu_{K-1}})$ | | Energy | $H(\sigma_{j_0}, \cdots, \sigma_{j_{K-1}})$ |

where $N$ is the number of spins in the $k$-th configuration, $\sigma_{j_k} = \pm 1$ are spin variables, and $L_{j_0 \ldots j_k}$ is an energy density function for $k$ nearest spins' configuration $(\sigma_{j_0}, \ldots, \sigma_{j_k})$.

**Modeling in Table 1**. Starting from an analogy between a state-action pair $\mu_k = (S_k, A_k)$ and a spin $j_k$, we can map an optimal policy $\pi^*(\mu_k) \in \{0, 1\}$ to the optimal single-qubit spin operator $\sigma_{j_k}^* \in \{-1, 1\}$ via $\pi^*(\mu_k) \longleftrightarrow (\mathbb{1}_{\mu_k} - \sigma_{\mu_k}^*)/2$, where $\pi(\mu_k)$ denotes the probability of taking action $A_k$ at state $S_k$, following policy $\pi$. The energy density function $L_{j_0 \ldots j_k}$ can be defined as the discounted reward on a path $(\mu_0, \cdots, \mu_{k-1})$ of length $k$,

$$L_{\mu_0, \ldots, \mu_k} = \gamma^k \cdot R(\mu_k) \cdot d_0(s_0) \cdot \prod_{\ell=0}^{k-1} \mathbb{P}(s_{\ell+1}|\mu_\ell), \quad \text{(obtained via Monte Carlo simulation)} \quad (2)$$

where $d_0(s_0)$ denotes the distribution of initial state $s_0$. Analogy to the quantum K-spin Ising model, we can derive an energy of a RL policy $H(\pi_{\mu_0}, ..., \pi_{\mu_{K-1}})$.

We formally express the objective of reinforcement learning (background is given in Appx. C) into a $K$-spin Hamiltonian equation (inspired by [20])

$$H(\theta) \triangleq -\mathbb{E}_{S_0, A_0}[Q^{\pi_\theta}(S_0, A_0)] = -\lim_{K \to \infty} \sum_{k=0}^{K-1} \sum_{\mu_0}^{\mathcal{S} \times \mathcal{A}} \cdots \sum_{\mu_k}^{\mathcal{S} \times \mathcal{A}} L_{\mu_0, \ldots, \mu_k} \pi_\theta(\mu_0) \cdots \pi_\theta(\mu_k), \quad (3)$$

where expectation is taken over $S_0 \sim d_0(\cdot)$, $A_0 \sim \pi_\theta(S_0, \cdot)$, and $L_{\mu_0, \ldots, \mu_k}$ is given in (2).

**Physical interpretation**: Analogy to a quantum K-spin system, $H(\theta)$ in (3) measures a random path's discounted reward (the "energy") without following any policy, and the Hamiltonian equation combinatorially enumerates all possible paths of length $K$ over the state-action space. The joint probability distribution, $\pi(\mu_0) \times \pi(\mu_1) \times \cdots \times \pi(\mu_{K-1})$, is decided by the policy $\pi$. monte carlo simulation The energy of a policy is a favorable criteria, since an optimal policy with minimum energy: 1). *achieves a relative high reward independent of the initialization*; and 2). *is robust to interference/noise in the inference stage*. In other words, the simulation process of the Hamiltonian term does not rely on any policy. Therefore, the Hamiltonian term is a suitable regularizer for both on-policy and off-policy algorithms.

**K-step truncation in practice**. Minimizing (3) is NP-hard [13]. Since $\gamma \in (0, 1)$, $\gamma^K$ monotonically decreases with look-ahead steps $K$, therefore, we truncate (3) to finite $K$ terms. One can show that these $K$ terms in (3) is a geometric sequence with a truncation error ratio $1 - \gamma^K$. Assuming $1 - \gamma^K \leq 1 - \epsilon$, where $\epsilon > 0$ is small, thus we have the look-ahead steps $K \geq \log_\gamma \epsilon$.

### 4.3 H-TERM HELPS CONVERGE TO A HIGH-QUALITY LOCAL MINIMA

We elaborate how adding the energy in (3) onto each state can help drive to the terminal state (a stationary policy), which fixes the issue in Section 1. We have $H(0) = 0$ for the terminal state 0.

- (a) **Shortest path problem (deterministic)**: $H(1) = -\sum_{k=1}^{\infty} b = -\infty$. At state 1, the Bellman's optimality equation becomes $V(1) = \max\{V(1) + \lambda H(1), b\}$. Independent of the initial value $V_0(1)$, an agent obtains a policy that always transits back to terminal state 0.

- (b) **Blackmailer's problem (stochastic)**: $H(1) = -\infty$. The Bellman's optimality equation becomes $V(1) = \max_a\{a + (1-a)(V(1) + \lambda H(1))\}$ for state 1. For any $V_0(1) < \infty$, the optimal policy becomes $a = 1$ that drives to the terminal state 0.

- (c) **Optimal stopping problem (terminating policies)**: any policy that takes infinite steps will have $H(x) = -\infty$, since at each step number $k$, there are always trajectories that jump to point 0 with reward $-c$; and a direct jumping policy will have $H(x) = -c$. Therefore, adding $H(x)$ to each point $x \neq 0$ will lead to a policy of *jumping back to point 0*.

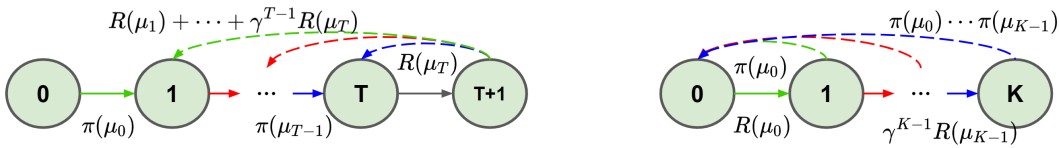

Figure 3: REINFORCE's policy gradient (left) VS. Hamiltonian's policy gradient (right).

## 5 STATIONARY DEEP REINFORCEMENT LEARNING

First, we propose a novel Hamiltonian policy gradient and the corresponding Monte Carlo estimator. Then, we present a stationary actor-critic algorithm with H-term.

### 5.1 HAMILTONIAN POLICY GRADIENT AND MONTE CARLO-BASED GRADIENT ESTIMATOR

We provide the policy gradient of the quantum K-spin Hamiltonian equation in (3), which are variants of the well-known policy gradient theorem [44]. We provide detailed derivations in Appx. E.

**Theorem 1.** (*Stochastic version*) *The Hamiltonian stochastic gradient of (3) is*

$$\nabla_\theta H(\theta) = -\mathbb{E}_{\mu_0,\ldots,\mu_{K-1}} \left[ \sum_{k=0}^{K-1} \gamma^k \cdot R(\mu_k) \cdot \nabla_\theta \log\left(\pi_\theta(\mu_0) \cdot \pi_\theta(\mu_1) \cdots \pi_\theta(\mu_k)\right) \right]. \quad (4)$$

Let $\eta_\theta(\cdot) : \mathcal{S} \to \mathcal{A}$ denote a deterministic policy, while we use $\widetilde{\pi}_{\theta,\delta}(\mu)$ to represent that a Gaussian noise (a.k.a, an exploration noise) with standard deviation $\delta > 0$ is added in the exploration process.

**Theorem 2.** (*Deterministic version*) *The Hamiltonian deterministic gradient of (3) is*

$$\nabla_\theta H'(\theta) = -\mathbb{E}_{\mu_0,\ldots,\mu_{K-1}} \left[ \sum_{k=0}^{K-1} \gamma^k \cdot R(\mu_k) \cdot \nabla_\theta \log\left(\widetilde{\pi}_{\theta,\delta}(\mu_0) \cdot \widetilde{\pi}_{\theta,\delta}(\mu_1) \cdots \widetilde{\pi}_{\theta,\delta}(\mu_k)\right) \right]. \quad (5)$$

The quantum K-spin Hamiltonian equation in (3) is a reformulation of (15). We verify the gradient calculation by showing that: when $K \to \infty$, the Hamiltonian stochastic and deterministic policy gradient $\nabla_\theta H(\theta)$ and $\nabla_\theta H'(\theta)$ are equal to the stochastic policy gradient $\nabla_\theta J(\theta)$ in [45] and deterministic policy gradient $\nabla_\theta J'(\theta)$ in [43], respectively.

Note that the gradient $\nabla_\theta H(\theta)$ in (4) and $\nabla_\theta H'(\theta)$ in (5) w.r.t. a distributional parameter $\theta$ takes an expectation form. Thus, a Monte Carlo gradient estimator is practically useful. We obtain the Monte Carlo gradient estimator of $\nabla_\theta H(\theta)$, illustrated in Fig. 3 (right), as follows

$$\nabla_\theta \widehat{H}(\theta) = -\frac{1}{N'} \sum_{i=1}^{N'} \left[ \sum_{k=0}^{K-1} \gamma^k \cdot R(\mu_k^i) \cdot \nabla_\theta \log\left[\pi_\theta(\mu_0^i) \cdots \pi_\theta(\mu_k^i)\right] \right]. \quad (6)$$

As a contrast, we provide the Monte Carlo gradient estimator of REINFORCE's [45] policy gradient, as illustrated in Fig. 3 (left), as follows

$$\nabla_\theta \widehat{J}(\theta) = \frac{1}{NT} \sum_{i=1}^{N} \left[ \sum_{t=0}^{T-1} G_t^i \cdot \nabla_\theta \log \pi_\theta(\mu_t^i) \right], \quad \text{where } G_t^i = \sum_{t'=t+1}^{T} \gamma^{t'-t-1} R(\mu_{t'}^i). \quad (7)$$

An interesting observation is that both gradient calculations follow a similar pattern as shown in Fig. 3. REINFORCE's policy gradient [45] in Fig. 3 (left) employs an estimate of future rewards, while Hamiltonian policy gradient in Fig. 3 (right) uses trajectories in replay buffer $\mathcal{D}_2$.

**Computational complexity**: we measure the computation complexity by the times of computing one $\nabla_\theta \log \pi_\theta(\mu)$. Assume $N = B$ and $N' = B'$, since most DRL algorithms use a mini-batch stochastic gradient decent methods. REINFORCE's [45] policy gradient in (7) takes $O(BT)$ computations, while Alg. 1 adds $O(B'K(K+1)/2)$ computations in each gradient update step, thus a total complexity of $O(BT + B'K(K+1)/2)$.

### 5.2 STATIONARY ACTOR-CRITIC ALGORITHM WITH H-TERM

Actor-critic algorithms in reinforcement learning perform a bilevel optimization, namely alternating between approximating a value function and optimizing a policy. In practice, a critic network with

---

**Algorithm 1** Stationary Actor-Critic Algorithm with H-term

---

1: **Input**: learning rate $\alpha$, temperature $\lambda$, look-ahead step $K$, and parameters $M, T, G, B, B'$
2: Initialize actor network $\pi$ and critic network $Q$ with parameters $\theta, \phi$, and replay buffers $\mathcal{D}_1, \mathcal{D}_2$
3: **for** episode $= 1, \cdots, M$ **do**
4:      Initialize state $s_0$
5:      **for** $t = 0, \cdots, T-1$ **do**
6:          Select action $a_t \sim \pi_\theta(\cdot|s_t)$
7:          Execute action $a_t$, receive reward $r_t$, and observe new state $s_{t+1}$
8:          Store a transition $(s_t, a_t, r_t, s_{t+1})$ in $\mathcal{D}_1$
9:      **end**
10:      Store a trajectory $\tau$ of length $T$ in $\mathcal{D}_2$
11:      **for** $g = 1, \cdots, G$ **do**
12:          Randomly sample a mini-batch of $B$ transitions $\{(s_i, a_i, r_i, s_{i+1})\}_{i=1}^{B}$ from $\mathcal{D}_1$
13:          Randomly sample a mini-batch of $B'$ trajectories (of length $K$) $\{\tau_j\}_{j=1}^{B'}$ from $\mathcal{D}_2$
14:          Update critic network using a conventional method
15:          Update actor network as $\theta \leftarrow \theta + \alpha \left( \nabla_\theta \widehat{J}(\theta) - \lambda \nabla_\theta \widehat{H}(\theta) \right)$.
16:      **end**
17: **end**

---

parameter $\phi$ approximates the $Q$-value function, and an actor network with parameter $\theta$ approximates the policy $\pi$, details given in Appx. G. However, since the critic's update is governed by the Bellman's optimality equation, actor-critic algorithms suffer the multiple fixed points problem.

Motivated by Section 4.3, we propose a novel H-term for both deterministic and stochastic actor-critic algorithms. Similar to the entropy term in [27], the proposed H-term is an add-on term to regularize the actor network and help it converge to a stationary policy. Specifically, the objective functions of actor and critic networks become:

$$\begin{cases} \text{Actor} : \max_\theta J_\pi(\theta, \phi) \triangleq (1-\gamma)\mathbb{E}_{S_0 \sim d_0, A_0 \sim \pi_\theta(S_0, \cdot)} \left[ Q_\phi(S_0, A_0) \right] - \lambda H(\theta), \\ \text{Critic} : \min_\phi J_Q(\theta, \phi) \triangleq \dfrac{1}{2}\mathbb{E}_{S \sim d_\theta(\cdot), A \sim \pi_\theta(S, \cdot)} \left[ (Q_\phi(S, A) - y(S, A))^2 \right], \end{cases} \quad (8)$$

where a target Q-value is $y(S_k, A_k) = R(S_k, A_k) + \gamma Q_\phi(S_{k+1}, A_{k+1})$, and $\lambda > 0$ is a temperature parameter. As an interpretation, the second term $-\lambda H(\theta)$ in the maximization objective function of actor network aims to find a minimum energy configuration for the MDP problem, namely, a policy $\pi$ that will add a minimum amount of energy to each state's value function (as in Section 4.3).

**New algorithm**. In Alg. 1, an agent interacts with an environment and alternatively updates its actor network and critic network. The algorithm has $M$ episodes and each episode consists of a (Monte Carlo) simulation process and a learning process (gradient estimation) as follows:

- During the (Monte Carlo) simulation process (lines 5-10 of Alg. 1), an agent takes action $a_t$ according to a policy $\pi_\theta(\cdot|s_t)$, $t = 0, \cdots, T-1$, generating a trajectory of $T$ steps/transitions. Then, these $T$ transitions are stored into a replay buffer $\mathcal{D}_1$, while the full trajectory $\tau = (s_0, a_0, r_0, s_1, \cdots, s_{T-1}, a_{T-1}, r_{T-1}, s_T)$ is stored in replay buffer $\mathcal{D}_2$.

- During the learning process ($G \geq 1$ updates in one episode) (lines 11-16 of Alg. 1), a mini-batch of $B$ transitions $\{(s_i, a_i, r_i, s_{i+1})\}_{i=1}^{B}$ and a mini-batch of $B'$ trajectories (of length $K$) $\{\tau_j = (s_0^j, a_0^j, r_0^j, s_1^j, \cdots, s_{K-1}^j, a_{K-1}^j, r_{K-1}^j, s_K^j)\}_{j=1}^{B'}$ are sampled from $\mathcal{D}_1$ and $\mathcal{D}_2$, respectively. The critic network is updated by a conventional method, e.g., minimizing the mean squared error (MSE) between an estimated Q-value and a target value. The actor is updated by a Monte Carlo gradient estimator over $B$ transitions and $B'$ trajectories.

**Two new hyperparameters**. We introduce two hyperparameters: a temperature $\lambda > 0$ that is a relative weight of the H-term, and a look-ahead step $K \leq T$ that defines the horizon of the H-term.

**Implementation of replay buffer $\mathcal{D}_2$**. After a full trajectory $\tau$ of length $T$ is generated, it is partitioned into $T - K + 1$ trajectories of length $K$. We rank them according to the cumulative reward and store the top portion, say $80\%$, into a new replay buffer $\mathcal{D}_2$ (line 10 of Alg. 1). We randomly sample a mini-batch of $B'$ trajectories from $\mathcal{D}_2$ (line 13 of Alg. 1) to compute the H-term.

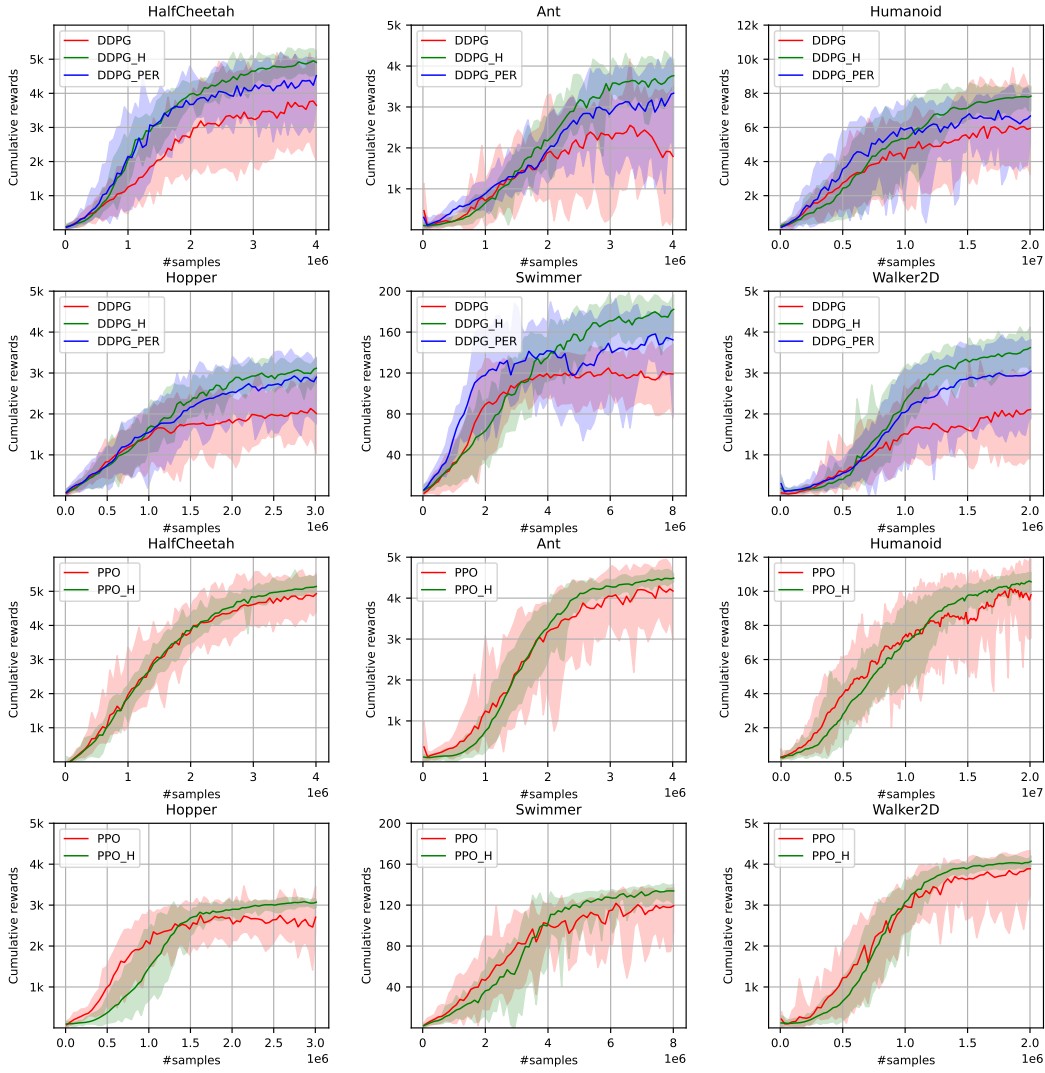

Figure 4: Cumulative rewards vs. #samples for compared DRL algorithms on six MuJoCo tasks.

# 6 Performance Evaluation

We evaluate the proposed H-term from four aspects: 1) converging to a high-quality local minima, 2) reducing variance, 3) driving to a stationary policy, and 4) the impact of trajectory length $K$. All experiments were executed on an NVIDIA DGX-2 server [12]. The server contains $8$ A100 GPUs, 320 GB GPU memory, and 128 CPU cores running at 2.25 GHz.

## 6.1 Experimental Settings

**Environments (tasks)**. We consider six challenging MuJoCo tasks [47], two combinatorial optimization tasks ( TSP and graph maxcut described in Appx. K), and two non-convex optimization tasks (MIMO beamforming in 5G/6G and non-convex deep learning classifier in Appx. L). For the MuJoCo tasks, the agent learns to control the locomotion of a robot and aims to move forward as quickly as possible. For graph maxcut and TSP, the agent learns to find a near-optimal solution. For MIMO beamforming and non-convex deep learning classifier, the agent learns to optimize the objective function. These tasks have high-dimensional continuous state space and action space, in which there exists multiple locally optimal polices as revealed in Section 3.1.

**Compared methods**. To evaluate both deterministic and stochastic algorithms, we choose Deep Deterministic Policy Gradient (DDPG) [34] and Proximal Policy Optimization (PPO) [42] for MuJoCo tasks. For TSP, graph maxcut, MIMO beamforming and non-convex deep learning classifier, we choose REINFORCE [45]. We implement the PPO algorithm with the GAE trick [41]. For a

Figure 5: Frequency of approx. ratio for a) TSP with $N = 100$, b) Graph maxcut with $N = 100$, c) MIMO beamforming with $N = 4$, and d) Non-convex deep learning classifier. A lower approximation ratio is better.

fair comparison, we keep the hyperparameters (listed in Appx. I) the same and make sure that the obtained results reproduce existing benchmark tests [16]. For combinatorial optimization, we use the same datasets from [14, 31].

**Performance metrics**. For MuJoCo tasks, we employ two performance metrics, the cumulative rewards and variance, while in Section 6.4, we further consider different policies and report the number of convergence. For TSP, graph maxcut, MIMO beamforming, and nonconvex deep learning classifier, we employ the approximation ratio $\max(\frac{Optimal}{Obj}, \frac{Obj}{Optimal})$ and its variance as the performance metric. We run each experiment with 20 random seeds and in each run we test 100 episodes.

## 6.2 H-TERM CONVERGES TO A HIGH-QUALITY LOCAL MINIMA

Experience replay is crucial in improving performance in terms of cumulative reward. The proposed H-term in (6) can be viewed as a novel experience replay technique for an actor network. Here, we add a compared algorithm, DDPG with Prioritized Experience Replay [40] (DDPG+PER), where PER prioritizes experience by the TD error to update a critic network.

In Fig. 4, both DDPG+PER and DDPG+H achieve a substantial improvement of cumulative reward. In particular, DDPG+H achieves the highest cumulative rewards in all six tasks, which are comparable to PPO's performance in Fig. 4. It is worthwhile to discuss the advantage of DDPG+H over DDPG+PER. DDPG+PER utilizes a prioritized replay strategy to obtain a more accurate critic network, however, it is updated via the Bellman equation with the trouble of multiple fixed points. In contrast, the H-term in DDPG+H is performed on the actor network. Our results indicate that an experience replay technique on actor network may be much more powerful.

In Fig. 5, REINFORCE+H improves the approximation ratio substantially. In particular, REIN-FORCE+H achieves an approximation ratio $\leq 1.05$ over 90% test cases in TSP, graph maxcut, MIMO beamforming, and non-convex deep learning classifier. Specifically, H-term achieves an approximation ratio 1.01 (near-optimal) over 90% tests on MIMO beamforming task. Our results indicate H-term helps the policies converge to high-quality local minimas over multiple runs.

## 6.3 H-TERM REDUCES VARIANCE

The PPO algorithm with GAE is regarded as the state-of-the-art algorithm in MuJoCo environments. However, it still has a very high variance (the shaded area) after the policies have converged, as shown in Fig. 4. We observe that, at the end of training, the PPO algorithm has a variance of 969.2, 1563.4, 2513.5, 905.3, 60.7, 1290.1 in the six tasks, respectively. Such a high variance is mainly due to the fact that the agent may converge to a random one of multiple policies.

In Fig. 4, the shaded areas of PPO+H ($K = 16$) are dramatically smaller, i.e., a variance of 228.4, 225.4, 683.7, 184.2, 31.6, and 296.8, respectively. The variance has been reduced by 65.2% $\sim$ 85.6%, which verifies the effectiveness of the proposed H-term. In Fig. 4, we also observe that the H-term can help the DDPG algorithm reduce variances, namely, the variances of DDPG+H are much smaller than those of vanilla DDPG and DDPG+PER. Therefore, we may conclude that the H-term guides the agent to search for a stationary policy among multiple feasible ones.

In Fig. 5, the variance of approximation ratios of REINFORCE+H ($K = 16$) are substantially smaller. The variance has been reduced by 60.16% $\sim$ 94.52%. The results indicate that the H-term guides policies to high-quality minimas over multiple runs.

Table 2: Experimental results on six challenging MuJoCo tasks.

| Tasks | Policies | PPO | | PPO+H ($K=8$) | | PPO+H ($K=16$) | | PPO+H ($K=24$) | |
|---|---|---|---|---|---|---|---|---|---|
| HalfCheetah | **running** | **13** | | **19** | | **20** | | **20** | |
| | flipping | 5 | 4720.8 | 0 | 5028.4 | 0 | 5104.3 | 0 | 4995.1 |
| | diving | 1 | ±969.2 | 1 | ±211.3 | 0 | ±228.4 | 0 | ±383.3 |
| | balancing | 1 | | 0 | | 0 | | 0 | |
| Ant | **running** | **17** | 4164 | **20** | 4505.3 | **20** | 4645.6 | **20** | 4662.5 |
| | jumping | 0 | ±1563.4 | 0 | ±253.6 | 0 | ±225.4 | 0 | ±277.5 |
| | flipping | 3 | | 0 | | 0 | | 0 | |
| Humanoid | **two-legs** | **7** | 9433.4 | **17** | 9670.3 | **16** | 10189.1 | **17** | 9942.2 |
| | one-leg | 12 | ±2513.5 | 3 | ±497.2 | 4 | ±683.7 | 3 | ±538.4 |
| | backward | 1 | | 0 | | 0 | | 0 | |
| Hopper | **hopping** | **10** | 2659.3 | **18** | 3116.5 | **20** | 3300.1 | **20** | 3340.7 |
| | diving | 8 | ±905.3 | 2 | ±289.4 | 0 | ±184.2 | 0 | ±191.5 |
| | balancing | 2 | | 0 | | 0 | | 0 | |
| Swimmer | **moving** | **14** | 110.7 | **20** | 130.6 | **20** | 132.5 | **19** | 132.2 |
| | balancing | 6 | ±60.7 | 0 | ±33.5 | 0 | ±31.6 | 1 | ±36.2 |
| Walker | **walking** | **5** | 5461.7 | **16** | 5819.9 | **16** | 5927.2 | **15** | 6089.3 |
| | diving | 8 | ±1290.1 | 2 | ±315.6 | 4 | ±296.8 | 5 | ±314.7 |
| | balancing | 7 | | 2 | | 0 | | 0 | |

More experimental results are given in Appx. I due to the space limit, including the cases of $K=8$ and $K=24$, and the H-term value during the training process. One may verify that the stationary policies have relative lower H-values.

### 6.4    H-TERM DRIVES TO PHYSICALLY STATIONARY POLICY

A key question needs to be answered: *is H-term really guiding the agent converge to a physically stationary policy?* Similar to Section 3.1, we perform observational experiments on MuJoCo tasks and measure the number of convergences to different policies over 20 runs. As shown in Table 2, the vanilla PPO algorithm converges to the physically stationary policy (**bold**) with 13, 17, 7, 10, 14, and 5 times for the six tasks, while the PPO+H ($K=16$) converges to the stationary policy with 20, 20, 16, 20, 20, and 16 times, respectively. From the empirical observation, we find that the PPO gets stuck in locally optimal policies, failing to find a consistent one. As expected, PPO+H can converge to the stationary policy with a substantially higher ratio, which verifies the effectiveness of the proposed H-term in finding a physically stationary policy.

### 6.5    IMPACT OF TRAJECTORY LENGTH $K$

We investigate the impact of trajectory length $K$. From (6), we know that a large $K$ means more accurate estimation of $\nabla_\theta H(\theta)$ but at a price of computations. Here, we evaluate PPO+H with $K=8,16$ and set the size of replay buffer $\mathcal{D}_2$ to $1,000$. In Table 2, we observe that the cumulative reward increases and the variance decreases as $K$ increases from 8 to 16. However, for the case $K=24$, both metrics get worse due to the out-of-memory issue and we reduce the replay buffer size to 800. The smaller replay buffer size hurts the diversity of the trajectories and may lead to a performance drop. Appx. I.2 provides results for replay buffer size 800.

## 7    CONCLUSIONS

In this paper, we have addressed the foundational issue of many local minimas. This issue leads to the instability of DRL algorithms, puts a challenge on their reliability and reproducibility, and thus limits the wider adoption of DRL algorithms in real-world tasks. As a fix of the problem, we propose a physically inspired regularizer by modeling a policy as a quantum K-spin Ising model. Experimental results show that the H-term helps DRL algorithms converge to a high-quality local minima, reduce the variance of cumulative rewards by $65.2\% \sim 85.6\%$ on six MuJoCo tasks, achieve an approximation ratio $\leq 1.05$ over $90\%$ test cases and reduce its variance by $60.16\% \sim 94.52\%$ on two combinatorial tasks and two non-convex optimization tasks, compared with those of existing algorithms over 20 runs, respectively.

For future works, we will explore the potential of directly training a policy network using (3) as in Appx. J, quantum simulator [29] and quantum reinforcement learning [6][19]. It is interesting to apply Monte Carlo estimator for unbiased policy gradient calculations. We would like to show that the proposed H-term can help distributional RL algorithms [4] find a stationary policy, since the distributional Bellman optimality operator is not a contraction and thus there is also no unique policy.

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
