# OpenReview forum: "Stationary Deep Reinforcement Learning with Quantum K-spin Hamiltonian Equation"
_ICLR.cc/2023/Conference — Submitted to ICLR 2023_

### Official Review · Reviewer_947x · 2022-10-23

**Confidence:** 3
**Correctness:** 2
**Technical Novelty And Significance:** 2
**Empirical Novelty And Significance:** 3
**Recommendation:** 3

**Clarity, Quality, Novelty And Reproducibility:**

This work studies a new regularization term for deep reinforcement learning. It is different from the existing regularization term, for instance, the entropy of the policy, which aims to enhance the exploration ability of the policy. It seems that the authors' strategy works well on several benchmarks such as MuJOCO. Therefore, I think this work is novel.

**Strength And Weaknesses:**

Pros:
- The presentation is easy to follow.
- The experiment results are comprehensive.
- The proposed method is easy to implement and has the potential to be applied to more tasks.

Cons:
- Some demonstrations are hard to understand. For instance, In Sec 4.3, I do not understand the Bellman equation used in (a-c). Why there does not exist the discount factor $\gamma$? If the authors want to discuss the case $\gamma=1$, they should not simply adopt the original Bellman equation, but the revised one for infinite-horizon MDPs, like (5) in [1].

[1] Jaksch, Thomas, Ronald Ortner, and Peter Auer. "Near-optimal Regret Bounds for Reinforcement Learning." Journal of Machine Learning Research 11 (2010): 1563-1600.

- Potential drawbacks of the proposed algorithm are not discussed. My understanding about why the proposed regularization term can help the policy have a smaller variance and better value is that it sacrifices the ability to explore. The regularization term $H$ forces the policy to follow the existing samples, especially the samples in the early steps $h\ll H$. Although such a strategy may work well on Mujoco dataset where the exploration is not crucial, I doubt such a strategy will work well on some harder instances, such as the combination lock example in [2].

[2] Agarwal, Alekh, et al. "Pc-pg: Policy cover directed exploration for provable policy gradient learning." Advances in neural information processing systems 33 (2020): 13399-13412.

**Summary Of The Paper:**

This paper studies a new regularization term for deep reinforcement learning which is inspired by the energy term from quantum physics. In general, the regularization term aims to maximize the adjusted likelihood function over the existing trajectories, which is opposite to the optimization goal of policy gradient which aims to maximize the expected reward in the future. Experiment results suggest that such an approach can improve the trained policy with better value and smaller variances.

**Summary Of The Review:**

This work proposes a new regularization term for policy optimization on deep reinforcement learning which seems a better choice than existing approaches on Mujoco dataset. However, I feel the demonstration of several examples in this paper is hard to understand, and the discussion of the limit of the proposed method is not well-discussed. Therefore, I recommend a reject to this paper.

---

> ### Author Response · Authors · 2022-11-19
> **# Response to Reviewer 947x**
>
> # Response to Reviewer 947x
>
> Thanks for your appreciation on our presentation, experiments, and the proposed method. The authors would like to give an detailed explanation on: 1) three demonstrative examples, and 2) the proposed H-term will not hurt the exploration ability.
>
>
> - *“Some demonstrations are hard to understand. For instance, In Sec 4.3, I do not understand the Bellman equation used in (a-c). Why there does not exist the discount factor γ? If the authors want to discuss the case γ=1, they should not simply adopt the original Bellman equation, but the revised one for infinite-horizon MDPs, like (5) in [1].”*
>
> Actually, the authors put the case $\gamma=1$ in Section 1 (Fig. 1) and Section 4 for easy understanding, while the discussion of the case $\gamma \in (0,1)$ is put in Appendix A.
>
> This paper focuses on the fundamental problem of multiple fixed points in Bellman equation. The authors thank for the interesting setting of online learnging [1] provided by the reviewer.
>
> H-term holds BOTH for the undiscounted case $\gamma=1$ and the discounted case $\gamma < 1$. The authors provide detailed explanations below.
>
> #### Case $\gamma = 1$
> **Shortest path problem (deterministic)** in Fig. 1a: At state $1$, an agent transits to either state $1$ or $0$ with reward $0$ or $b$, respectively. Assume the value function for state $0$ is $V(0) = 0$. The Bellman's optimality equation for state $1$ is $V(1) = \max\{V(1), b\}$, where any $V(1) \geq b$ is a feasible solution. If initialize $V_0(1) \geq b$, a resulting policy is that an agent at state $1$ always transits back to state $1$; otherwise, drives to terminal state $0$ (always returns back to itself with reward $0$).
>
> Then, we elaborate how the proposed H-term fixes the problem.
>
>  $H(1) = - \sum_{k=1}^{\infty} b = - \infty$. At state $1$, the Bellman's optimality equation becomes $V(1) = \max\{V(1) + \lambda H(1), b\}$. Independent of the initial value $V_0(1)$, an agent obtains a policy that always transits back to terminal state $0$.
>
> **Blackmailer's problem (stochastic)** in Fig. 1b: At state $1$, a profit maximizing blackmailer demands a cash amount  $a \in (0, 1]$; a victim transits to state $1$ with probability $a$ or state $0$ with probability $1-a$, respectively. At state $0$, a victim always refuses to yield, i.e., $V(0)=0$. The Bellman's optimality equation for state $1$ is $V(1) = \max_a \{a + (1-a)V(1)\}$, where any $V(1) \geq 1$ is a feasible solution. If initialize $V_0(1) > 1$, the blackmailer's policy is demanding $a \rightarrow 0$ to keep the victim at state $1$; otherwise, demanding $a = 1$ that drives the victim to terminal state $0$.
>
> Then, we elaborate how the proposed H-term fixes the problem.
>
>  $H(1) = - \infty$. The Bellman's optimality equation becomes $V(1) = \max_a \{a + (1-a)(V(1) + \lambda H(1))\}$ for state $1$. For any $V_0(1) < \infty$, the optimal policy becomes $a=1$ that drives to the terminal state $0$.
>
> **Optimal stopping problem (terminating policies)** in Fig. 1c: In a space $\mathbb{R}^2$ with terminal state of point $0$, an agent at point $x \neq 0$ moves to either point $0$ with negative reward $-c$ or point $\alpha x$ with reward $- ||x||$, respectively, where $\alpha \in (0, 1)$. The Bellman's optimality equation is $V(x) = \max \{ - c, - ||x|| + V(\alpha x)\}$ and the optimal policy is to continue inside the sphere of radius $(1- \alpha)c$ and to stop outside. If add a cone region $C$ within which an agent always receives a reward $-c$, a second policy is jumping to point $0$ at any point in region $C$.
>
> Then, we elaborate how the proposed H-term fixes the problem.
>
> any policy that takes infinite steps will have $H(x) = -\infty$, since at each step number $k$, there are always trajectories that jump to point $0$ with reward $-c$; and a direct jumping policy will have $H(x) = -c$. Therefore, adding $H(x)$ to each point $x \neq 0$ will lead to a policy of jumping back to point $0$.

---

> > ### Author Response · Authors · 2022-11-19
> > **Response to Reviewer 947x (2/3)**
> >
> > #### Case $\gamma \in (0,1)$
> > **Shortest path problem (deterministic, $\color{red}{discounted~ case}$)**: Given two states $1$ and $0$, an agent at state $1$ transits to either state $1$ or $0$ with rewards \color{red}{$r = c$} or $r = b$, respectively. $\color{red}{c > (1 - \gamma) \cdot b}$. At state $0$, the value function is $V(0) = 0$. At state $1$, the Bellman's optimality equation is $V(1)=\max\{c + \color{red}{\gamma}\cdot V(1), b\}$, where any $V(1) \geq  \color{red}{(b-c) / \gamma}$ is a solution. If initialize $V_0(1) \geq b$, an agent obtains a policy that always transits back to state $1$; otherwise, a result policy drives to terminal state $0$.
> >
> > Then, we elaborate how the proposed H-term fixes the problem.
> >
> > Assume $V_0(1)\geq b$ and $c> (1-\gamma)b$, we have
> > \begin{equation}
> > \begin{split}
> >     &V_1(1) = c + \gamma \cdot V_0(1) \geq  c + \gamma\cdot b  > b \\
> >     &V_2(1) = c + \gamma \cdot c + \gamma^2 \cdot V_0(1) \geq (1 + \gamma)c + \gamma^2 b > b\\
> >     &V_3(1) = c + \gamma \cdot c + \gamma^2 c+\gamma^3 \cdot V_0(1) \geq (1 + \gamma+\gamma^2)c + \gamma^3 b > b\\
> >     &\cdots\\
> >     &V_k(1) = \sum_{i= 0}^{k - 1}\gamma^i \cdot c + \gamma^k \cdot V_0(1) \geq \sum_{i= 0}^{k - 1}\gamma^i \cdot c + \gamma^k b > b\\
> >     &\cdots\\
> >     &V^*(1) = \sum_{i = 0}^{\infty} \gamma^i\cdot c = \frac{1}{1 - \gamma}c > b\\
> > \end{split}
> > \end{equation}
> > The values of $H(0)$ and $H(1)$ are as follows:
> > \begin{equation}
> > \begin{split}
> > H(0)=0,H(1) = -b - \sum{k = 2}^\infty( \sum{i = 1}^{k - 1}\gamma^{i-1}\cdot c + \gamma^k b) = -\infty.
> > \end{split}
> > \end{equation}
> > Adding the above H-values to state $1$ and $0$, respectively, we have
> > \begin{equation}
> >     \begin{split}
> >         V^*(1) + H(1) &= \sum_{i = 0}^\infty\gamma^i\cdot c  -\infty  = -\infty\\
> >         V^*(0) + H(0) &= b.
> >     \end{split}
> > \end{equation}
> > Therefore, $V^*(1) + H(1) < V^*(0) + H(0)$, independent of the initial value $V_0(1)$. That is, an agent always obtains a policy that drives to terminal state $0$ at step 1.
> >
> > **Blackmailer's problem (stochastic, $\color{red}{discounted~ case}$)**: Different from (a), a profit maximizing blackmailer/agent at $1$ demands a cash amount $a \in (0, 1]$ (an action), while a victim transits to state $1$ with probability $a$ or to state $0$ with probability $1-a$, respectively. At state $0$, a victim always refuses to yield to the blackmailer's demand, i.e., $V(0)=0$. The Bellman's optimality equation is $V(1) = \max_a \{a + \color{red}{\gamma}\cdot(1-a)V(1)\}$ for state $1$, where any $V(1) \geq 1$ is a feasible solution. If initialize $V_0(1) = c > 1$, the blackmailer's policy is demanding $a \rightarrow 0$ at the $k$-th step to keep the victim stay at state $1$, for any $\color{red}{k \leq K_0 = -\lfloor\log_{\gamma}c\rfloor}$; and taking $\color{red}{a = 1}$ to transit to terminal state $0$ at the $k$-th step, for any $\color{red}{k \geq K_0 + 1}$; otherwise initialize $V_0(1) = c \leq 1$, the result policy is demanding the maximum $a = 1$ that drives the victim to a refusal state $0$ (a terminal state).
> >
> > Then, we elaborate how the proposed H-term fixes the problem.
> >
> > If initialize $V_0(1) = c > 1$, the blackmailer's policy is demanding $a \rightarrow 0$ at the $k$-th step to keep the victim stay at state $1$, for any $\color{red}{k \leq K_0 = -\lfloor\log_{\gamma}c\rfloor}$; and taking $\color{red}{a = 1}$ to transit to terminal state $0$ at the $k$-th step, for any $\color{red}{k \geq K_0 + 1}$; otherwise initialize $V_0(1) = c \leq 1$, the result policy is demanding the maximum $a = 1$ that drives the victim to a refusal state $0$ (a terminal state).
> >
> > The values of $H(0)$ and $H(1)$ are as follows:
> > \begin{equation}
> >     H(0) = 0, H(1) = -\sum_{k = 1}^{\infty}\sum_{i = 1}^{k - 1}\gamma^{i - 1}\cdot a= -\infty.
> > \end{equation}
> > For arbitrary initial value of $V_0(1)$, $V_1(1) = a + (1 - a) \cdot \gamma (V_0(1) + H(1))$ take maximum $V_1(1) = 1$ when $a = 1$. Therefore, the policy always drives to terminal state $0$ at step $1$.
> >
> > **Optimal stopping problem (terminating policies, $\color{red}{discounted~case})$**: In a space $R^2$ with terminating state at point $0$, at point $x \neq 0$ an agent moves to either point $0$ with negative reward $-c$ or point $\alpha x$ with reward $- ||x||$, respectively, where $\alpha \in (0, 1)$. The Bellman's optimality equation is $V(x) = \max \{ - c, - ||x|| +\color{red}{\gamma}\cdot V(\alpha x)\}$ and the optimal policy is to continue inside the sphere of radius $(1- \alpha)c$ and to stop outside. If add a cone region $C$ within which an agent always receives a reward $-c$, a second policy is jumping to point $0$ at any point in region $C$.
> >
> > Then, we elaborate how the proposed H-term fixes the problem.
> >
> > Any policy that takes infinite steps will have
> > \begin{equation}
> > H(x) = -c -\sum_{k = 2}^{\infty}\left[\sum_{i = 1}^{k - 1}\gamma^i\cdot\alpha^i\cdot \|x\| + \gamma^k\cdot(-c)\right]=  -\infty
> > \end{equation}
> > and a direct jumping policy will have $H(x) = -c$.

---

> > > ### Author Response · Authors · 2022-11-19
> > > **Response to Reviewer 947x (3/3)**
> > >
> > > - *“the drawback of sacrificing the ability to explore”*
> > >
> > > The authors understand that the reviewer concerns about the ability of exploration. Firstly, H-term as a regularizer has a weight coefficient $\lambda$ that controls the ability of exploration. We can set $\lambda = t / T$, where $T$ is the total training epochs, $t$ is the current epoch. $\lambda$ will start with a small value. Hence, the ability to explore will not be sacrificed.
> > >
> > > Secondly, given the setting of combinatorial problems (e.g., TSP and graph max-cut), agents do not get a reward until reaching the terminate state. It is a sparse reward setting. In Fig. 5, H-term helps converge to a better local minima. For hard-to-explore or sparse reward tasks, e.g., the Montezuma’s Revenge, there is no evidence (in hundreds of our testings) that H-term is incompatible with techniques like intrinsic reward. Thanks for pointing it out.

---

### Official Review · Reviewer_arYu · 2022-10-24

**Confidence:** 4
**Correctness:** 2
**Technical Novelty And Significance:** 1
**Empirical Novelty And Significance:** 2
**Recommendation:** 3

**Clarity, Quality, Novelty And Reproducibility:**

The paper is overall poorly written and hard to follow. Meanwhile, I believe the proposed method is not so novel.

**Strength And Weaknesses:**

### Weaknesses:
* I feel the motivation for the paper is not so clear. I don’t see any superiority to reinterpret the reinforcement learning problem in this K-spin Ising Model framework. It only covers a very small subset of the Markov Decision Process problem (e.g. the optimal action should be on a lattice), and there are no quantum related things here.
* I think the authors do not have a clear understanding of statistical mechanics. At least, the term “Hamiltonian equations” is created by the authors themselves. We only have Hamilton’s equations of motions (or Hamiltonian mechanics). In fact, the authors only use the Hamiltonian of the K-spin Ising model. I did expect there are some more motivations from statistical mechanics, e.g. we need to develop specific update rules to reserve the Hamiltonian for certain purposes. However, if I understand correctly, what the authors do do not have any relations with this K-spin Ising Model framework.
* This new policy gradient estimator seems closely related to other estimators, as $\nabla_{\theta} \log (\Pi_i \pi_{\theta}(a_i |s_i)) = \sum_i \nabla_{\theta} \log \pi_{\theta} (a_i|s_i)$. I even suspect by this decomposition the proposed policy gradient estimator is identical to the original one and the only difference between the newly proposed estimator and the original policy gradient estimator is how to estimate the cumulative reward (by the critic or by roll-out). The authors should provide more discussions on this.


**Summary Of The Paper:**

This paper introduces a new policy gradient method, which can potentially reduce the variance of the policy gradient estimator.

**Summary Of The Review:**

I don’t think the newly proposed policy gradient estimator has any connections to the K-spin Ising Model. Meanwhile, I suspect the newly proposed policy gradient estimator is identical to the standard one and the only difference is in the empirical estimation of the cumulative reward.

---

> ### Author Response · Authors · 2022-11-19
> **Response to Reviewer arYu (1/2)**
>
> #  Response to Reviewer arYu (1/2)
>
> Thanks for your comments on our paper. The authors would like to provide responses to the following points 1) the motivation of the proposed K-spin Ising model, 2) the improper use of *Hamiltonian equation*, and 3) the relationship between the Hamiltonian policy gradient and the original policy gradient.
>
> - *“I feel the motivation for the paper is not so clear. I don’t see any superiority to reinterpret the reinforcement learning problem in this K-spin Ising Model framework. It only covers a very small subset of the Markov Decision Process problem (e.g. the optimal action should be on a lattice), and there are no quantum related things here.”*
>
> The authors would like to clarify how we model the policy as a quantum K-spin Ising model. The K-spin Ising model is not the lattice Ising model in statistical mechanics. The *“lattice”* Ising model is used for analyzing the phase transition phenomena. Take the instance of phase transitions in magnetic fields, the energy of a configuration of the magnetic spins is given by the Hamiltonian function, $$H(\sigma) = -\sum_{<i, j>}L_{i,j}\sigma_i\sigma_j,$$ where the sum $\sum_{<i, j>}L_{i,j}\sigma_i\sigma_j$ is over pairs of adjacent spins, the notation $<i,j>$ denotes spin $i$ and spin $j$ are nearest neighbours.
> For Hamiltonian of quantum K-spin Ising model, we have $$H(\sigma_{j_0}, \cdots, \sigma_{j_{K-1}}) = -\sum_{k=0}^{K-1}\sum_{j_1}^N\cdots\sum_{j_k}^NL_{j_0\cdots j_{k-1}}\sigma_{j_0}\cdots\sigma_{j_{k-1}},$$ where $N$ is the number of spins in the $k$-th configuration, $\sigma_{j_i}=\pm1$ are spin variables, and $L_{j_0\cdots j_k}$ is an energy density for $k$-nearest spins' configuration $(\sigma_{j_0},\cdots, \sigma_{j_{k}})$.
>
> Below the authors would like to take the opportunity to clarify our analogy between the K-spin Hamiltonian and the policy.
>
> Starting from an analogy between a state-action pair $\mu_k = (S_k, A_k)$ and a spin $j_k$, we can map an optimal policy $\pi^*(\mu_k) \in \{0, 1\}$ to the optimal single-qubit spin operator $\sigma^*_{j_k} \in \{-1, 1\}$ via  $\pi^*(\mu_k) \longleftrightarrow ( 1_{\mu_k} - \sigma^*_{\mu_k} )/2$, where $\pi(\mu_k)$ denotes the probability of taking action $A_k$ at state $S_k$, following policy $\pi$. The energy density function $L_{j_{0} \ldots j_{k}}$ can be defined as the discounted reward on a path $(\mu_0, \cdots, \mu_{k-1})$ of length $k$,
>     $$ L_{\mu_0, ..., \mu_k} = \gamma^k\cdot R(\mu_k)\cdot d_0(s_0) \cdot \prod_{\ell = 0}^{k - 1} \mathbb{P}(s_{\ell + 1}|\mu_{\ell}),~~~\text{(obtained via Monte Carlo simulation)}$$
>     where $d_0(s_0)$ denotes the distribution of initial state $s_0$. Analogy to the quantum K-spin Ising model, we can derive an energy of a RL policy $H(\pi_{\mu_0}, ..., \pi_{\mu_{K-1}})$.
>
> To avoid misunderstanding, the authors would like to explain in more details. The Ising model is a universal model, e.g. NP-hard problems [2], and iterative optimization algorithms [3]. Both optimal policy $\pi^* \in \{0, 1\}$ and policy $\pi \in [0, 1]$ could be naturally mapped to a quantum field (a spin configuration). In physics, a spin orients at an angle in $[0, 2\pi)$ and takes continuous value in $[-1, 1]$, while the optimal spin configuration takes discrete value $\{-1, 1\}$. Therefore, the optimal policy $\pi^*$ corresponds to the optimal spin configuration, and the policy $\pi$ corresponds to the case when spins take continuous values.
> [1] Fan Y 2011 One-dimensional Ising model with k-spin interactions Eur. J. Phys. 32 1643–50
> [2] Lucas, Andrew. "Ising formulations of many NP problems." Frontiers in physics (2014): 5.
> [3] Li, Ke, and Jitendra Malik. "Learning to Optimize." ICLR, 2017.
>
> - *“I think the authors do not have a clear understanding of statistical mechanics. At least, the term “Hamiltonian equations” is created by the authors themselves. We only have Hamilton’s equations of motions (or Hamiltonian mechanics). In fact, the authors only use the Hamiltonian of the K-spin Ising model. I did expect there are some more motivations from statistical mechanics, e.g. we need to develop specific update rules to reserve the Hamiltonian for certain purposes.”*
>
> The authors agree that the use of "Hamiltonian equation" is not appropriate. The use of "Hamiltonian" as an energy measure is more accurate. The authors take a quantum perspective by modeling a policy as a K-spin Ising model and employ a Hamiltonian to measure the energy of a policy.

---

> > ### Author Response · Authors · 2022-11-19
> > **Response to Reviewer (2/2)**
> >
> > - *“This new policy gradient estimator seems closely related to other estimators, as $\nabla_\theta(\prod_i\pi_\theta(a_i|s_i)) = \sum_i\nabla_\theta\log(\pi_\theta(a_i|s_i))$. I even suspect by this decomposition the proposed policy gradient estimator is identical to the original one the only difference between the newly proposed estimator and the original policy gradient estimator is how to estimate the cumulative reward (by the critic or by roll-out).”*
> >
> > The authors agree that the reviewer would like to clearly see the relationship between the Hamiltonian policy gradient and the REINFORCE's policy gradient. The Hamiltonian policy gradient is a variant of the orginal policy gradient. The original policy gradient is with respect to the cumulative reward function (the expectation is taken over trajectories); the Hamiltonian policy gradient is derived from the expectation of discounted rewards along a trajectory. The Hamiltonian policy gradient is equivalent to the original policy gradient when $K\rightarrow +\infty$.
> >
> >
> > For REINFORCE's policy gradient, $$\nabla J(\theta) = E_{s\sim d_\theta, a\sim\pi_\theta}[G_t\nabla_\theta \log(\pi_\theta(s,a))]$$ one uses the Monte Carlo method by sampling $N$ trajectories and updates the network parameters using Eqn. (7). Note that $G_t = \sum_{t'=t+1}^T\gamma^{t'-t-1} R(\mu_{t'}), t = 0, 1,\cdots, T-1$ is the discounted cumulative rewards along one trajectory.
> >
> > For our Hamiltonian policy gradient, $$\nabla_{\theta} H({\theta}) =-E_{\mu_0,..., \mu_{K-1}}\left[\sum_{k = 0}^{K-1} \gamma^k\cdot R(\mu_k)\cdot\nabla_\theta \log \left(\pi_\theta(\mu_0)\cdot\pi_\theta(\mu_1)\cdots\pi_{\theta}(\mu_k)\right)\right],$$ we use the Monte Carlo method by sampling $N'$ trajectories with length $K$ and update the network parameters using Eqn. (6).
> >
> > REINFORCE's policy gradient essentially estimates the expectation of future rewards, while our Hamiltonian policy gradient uses the trajectories with length $K$ sampled from replay buffer.
> >
> > Let us consider a simple example: $T = 10, K = 3, N' = N = 10$.
> >
> > REINFORCE's policy gradient:
> > 1. Compute $G_t^i,~~~~t = 0, 1, \cdots, 9, ~~~~ i= 1, \cdots, 10$.
> > 2. Update the network parameters with $\frac{1}{10}\frac{1}{10}\sum_{i = 1}^{10}\left[\sum_{t=0}^{9} G_t^i\cdot  \nabla_\theta\log\pi_\theta(\mu^{i}_t)\right].$
> >
> > Hamiltonian policy gradient:
> > 1. Compute $\sum_{k=0}^{2}\gamma^k\cdot R(\mu_{t_i+k})\log[\pi_\theta(\mu_{t_i + 0})\cdots\pi_\theta(\mu_{t_i+k})],~~~~ i = 1, 2, \cdots, 10.$
> > 2. Update the network parameters with $- \frac{1}{10}\sum_{i = 1}^{10} \left[\sum_{k = 0}^{2} \gamma^k\cdot R(\mu^i_k)\cdot  \nabla_\theta\log\left[\pi_\theta(\mu^i_0)\cdots\pi_\theta(\mu^i_k)\right]\right]$.
> >
> > (In our experiments in Section 6, we consider T = 1000; K = 8, 16, and 24; N = N' = 64)

---

### Official Review · Reviewer_6uJh · 2022-11-02

**Confidence:** 3
**Correctness:** 2
**Technical Novelty And Significance:** 2
**Empirical Novelty And Significance:** 3
**Recommendation:** 5

**Clarity, Quality, Novelty And Reproducibility:**

The writing and notations of the paper need to be heavily polished. There are lots of misleading notations in the current version. For example, in Eq. (3) $H(\theta)$ the $\theta$ is parameters of the policy, while in the above paragraph the argument of $H$ is the chain of policies, and later in Sec. 4.3 the arguments of $H$ become the state indices. In Sec. 5.1 computational complexity, does the ‘computations’ here actually mean the gradient computation through the whole network in the backpropagation? It’s very vague to say the number of computations.

In Section 6.1 the performance metrics, the ‘Optimal’ and ‘obj’ are not explained.

The idea is novel.

The code is given and should be reproducible.


**Strength And Weaknesses:**

Strength:

Stabilizing the training of DRL is an important topic. The idea from Ising model for minimizing the energy in policy optimization is novel.

Weakness:

The relationship of the Hamiltonian policy gradient (HPG) and the REINFORCE policy gradient (RPG) is less clear. From the method description it looks like the HPG is an alternative format for estimating the gradient of objective in RPG, however in the end the H-term is only a regularizer.

Since the variance estimation is important in the paper, I would also suggest to report the results with stratified bootstrap confidence intervals [1].

The proof of variance reduction using H-term is not convincing. In appendix F.2 the paper gives a theoretical justification of the variance reduction optimizing the policy with an additional H-term. However, it looks like the proof is merely about showing the strong correlation of $H$ and $J$ thus reducing the total variance when $\nabla_J(\theta)$ is subtracted by $\lambda \nabla H(\theta)$ with a small $\lambda$. If this is the case, simply letting $H=J$ with $\lambda<1$ can reduce the variance, but with no help to the policy gradient optimization.

K-step truncation is less clear. In the paragraph in Sec. 4.2 about truncation of K, what is the truncation error about and what is $\epsilon$ here, and why does it simply want $\gamma^K\ge \epsilon$?

Typos:
Line 631 in appendix should refer to `Section 4.2`
Paragraph Physical interpretation in Section 4.2, ‘monte carlo simulation’.


[1] Agarwal, Rishabh, et al. "Deep reinforcement learning at the edge of the statistical precipice." Advances in neural information processing systems 34 (2021): 29304-29320.


**Summary Of The Paper:**

The paper proposes a regularization term (namely H-term) for stabilizing the training of DRL. The inspiration of the H-terms comes from the K-spin Ising model with energy minimization principle. Experiments on PPO and DDPG for MuJoCo show variance reduction of the proposed method.

**Summary Of The Review:**

The paper investigates an interesting problem about stabilizing DRL training. But the proposed H-term is not well justified from a theoretical perspective. The performance improvement of different MuJoCo tasks seem to be marginal. Experiments testify the reduced variances of policy performance but not policy gradient estimation. I would also suggest showing clear evidence of variance reduction of the gradients on small games with the ground truth gradient available.

---

> ### Author Response · Authors · 2022-11-17
> **Response to Reviewer 6uJh (1/3)**
>
> Thanks for your appreciation on the motivation. The authors provide responses to the following points: 1) the relationship between the Hamiltonian policy gradient and REINFORCE policy gradient, 2) the proof of variance reduction, 3) truncation of K, 4) the notation of H, and 5) the computation complexity.
>
> - *“The relationship of the Hamiltonian policy gradient (HPG) and the REINFORCE policy gradient (RPG) is less clear. ”*
>
> The authors agree that readers would like to clearly see the relationship between the Hamiltonian policy gradient and the REINFORCE policy gradient. It is a key contribution that is an add-on term in Alg. 1.
>
> For REINFORCE's policy gradient,
> $$\nabla J(\theta) =  E_{s \sim d_\theta, a \sim \pi_\theta}[G_t \nabla_\theta \log(\pi_\theta(s, a))],$$
> one uses the Monte Carlo method by sampling $N$ trajectories and updates the network parameters using Eqn (7).
> Note that $G_t = \sum_{t'=t+1}^T \gamma^{t'-t-1} R(\mu_{t'}), t = 0, 1,\cdots, T-1$ is the discounted cumulative rewards along one trajectory.
>
> For our Hamiltonian policy gradient, $$\nabla_{\theta} H({\theta}) =-E_{\mu_0,..., \mu_{K-1}}\left[\sum_{k = 0}^{K-1} \gamma^k\cdot R(\mu_k)\cdot\nabla_\theta \log \left(\pi_\theta(\mu_0)\cdot\pi_\theta(\mu_1)\cdots\pi_{\theta}(\mu_k)\right)\right],$$ we use the Monte Carlo method by sampling $N'$ trajectories with length $K$ and update the network parameters using Eqn (6).
>
> REINFORCE policy gradient essentially estimates the expectation of future rewards, while our Hamiltonian policy gradient uses the trajectories with length $K$ sampled from replay buffer.
>
> Let us consider a simple example: $T = 10, K = 3, N' = N = 10$.
>
> REINFORCE's policy gradient:
> 1. Compute $G_t^i,~~~~t = 0, 1, \cdots, 9, ~~~~ i= 1, \cdots, 10$.
> 2. Update the network parameters with $\frac{1}{10}\frac{1}{10}\sum_{i = 1}^{10}\left[\sum_{t=0}^{9} G_t^i\cdot  \nabla_\theta\log\pi_\theta(\mu^{i}_t)\right]$
>
> Hamiltonian policy gradient:
> 1. Compute $\sum_{k=0}^{2}\gamma^k\cdot R(\mu_{t_i+k})\log[\pi_\theta(\mu_{t_i + 0})\cdots\pi_\theta(\mu_{t_i+k})],~~~~ i = 1, 2, \cdots, 10$
> 2. Update the network parameters with $- \frac{1}{10}\sum_{i = 1}^{10} \left[\sum_{k = 0}^{2} \gamma^k\cdot R(\mu^i_k)\cdot  \nabla_\theta\log\left[\pi_\theta(\mu^i_0)\cdots\pi_\theta(\mu^i_k)\right]\right]$.
>     (In our experiments in Section 6, we consider T = 1000; K = 8, 16, and 24; N = N' = 64)
>
> The authors would like to take this opportunity to state our contributions:
> We take a quantum perspective by using the K-spin Ising model to model the policy. The Hamiltonian policy gradient is a variant of the original policy gradient. The original policy gradient is with respect to the cumulative reward function (the expectation is taken over trajectories); the Hamiltonian policy gradient is derived from the expectation of discounted rewards along a trajectory. The Hamiltonian policy gradient is quivalent to the orginal policy gradient when $K\rightarrow +\infty$.
>
> - *“From the method description it looks like the HPG is an alternative format for estimating the gradient of objective in RPG, however in the end the H-term is only a regularizer.”*
>
> Actually, the Hamiltonian policy graident outperforms REINFORCE'S policy gradient on the Frozenlake task and Girdworld task, as shown in Appendix J, Fig. 9. For harder tasks, the computation of the H-term is compute-intensive and NP-hard. Our "know-how" is to employ the H-term as a regularizer. From the experimental results, the H-term as a regularizer reduces the traning variances dramatically. The H-term guides the policy to high-quality local minimas. The authors re-describe the reasons below:
>
> 1. The combinatorial optimization tasks and non-convex optimization tasks have the problem of many local minimas. The problem of many local minimas is worsened by the multiple fixed points problem of Bellman equation. In Fig. 1 and Section 3.1, the authors has demonstrated the problems of many local minimas in combinatorial optimization tasks, non-convex optimization tasks, and the Bellman equation.
>
> 2. DRL algorithms fail to find a high-quality local minima from many local minimas. Fig. 2 shows that DRL algorithms converge to different policies.
>
> 3. From the perspective of physics, the H-term as a regularizer helps the policy converge to a high-quality local minima with minimum energy. An optimal policy with minimum energy achieves a relatively high reward independent of the initialization and is robust to interference/noise in the inference stage.
>
> 4. Furthermore, both the Ising model and the lowest-energy state are fundamental in physics. The authors are quite impressed by the fact that the Hamiltonian equation (simple and easy to implement) can be used as an add-on term to most actor-critic DRL algorithms (note that we tested over 5 algorithms, i.e., DDPG, PPO, SAC, TD3, REINFORCE), and such an add-on term effectively addresses the “instability” of DRL.

---

> > ### Author Response · Authors · 2022-11-17
> > **Response to Reviewer 6uJh (2/3)**
> >
> >
> >
> > - *“Since the variance estimation is important in the paper, I would also suggest to report the results with stratified bootstrap confidence intervals [1]. The proof of variance reduction using H-term is not convincing. In appendix F.2 the paper gives a theoretical justification of the variance reduction optimizing the policy with an additional H-term. However, it looks like the proof is merely about showing the strong correlation of H and J thus reducing the total variance when ∇J(θ) is subtracted by λ∇H(θ) with a small λ. If this is the case, simply letting H=J with λ<1 can reduce the variance, but with no help to the policy gradient optimization.”*
> >
> >
> >     The authors agree that the proof of variance reduction shoule be more convincing. However, this is not a theoretical paper. The authors take a quantum perspective by modeling a policy as a K-spin Ising model and employ a Hamiltonian equation to measure the energy of a policy. The proposed K-spin Hamiltonian regularization term guides the policy to converge to a high quality local minimas. The experimental results show that the H-term reduces the training variances.
> >
> >     The authors would like to present the following points about the variance reduction, 1) the counter example of replacing $\lambda H(\theta)$ with $\lambda J(\theta)$ presented by the reivewer is not proper, 2) the proposed H-term is inspired by quantum K-spin Ising model, 3) H-term guides the policies converge to high-quality local minimas from multiple local minimas as shown in the three examples and the experimental results.
> >
> >     1) The reviewer gives a counter example of the proof of variance reduction: replacing $\lambda H(\theta)$ with $\lambda J(\theta)$ does not reduce the variance. The authors would like to point out the counter exaxmple is not proper:
> >
> >         The variance reduction of Monte Carlo estimator is by subtracting a baseline term [1]: $E_{p(x;\theta)}[(f(x) - \beta)\nabla_\theta\log p(x;\theta)]$, where $\beta$ is a constant, as shown in Eqn (39). In the REINFORCE's policy gradient, ones have
> >         $$E_{s\sim d_{\theta}, a\sim \pi_\theta}[G_t\nabla_\theta\log \pi_\theta(s,a)],$$ where $G_t = \sum_{t'=t+1}^T\gamma^{t'-t-1} R(\mu_{t'})$. The computation of $G_t$ is related to future rewards $R_{t'+1} \cdots R_T$. It means $G_t$ is a function of the policy $\pi_\theta$. Hence, replacing $\lambda H(\theta)$ with $\lambda J(\theta)$ violates the condition that $\beta$ is a constant.
> >
> >     2) The proposed H-term is inspired by physics. Namely, an optimal policy with minimum energy achieves a relatively high reward independent of the initialization and is robust to interference/noise in the inference stage. The authors take a quantum perspective by modeling a policy as a K-spin Ising model and employ a Hamiltonian equation to measure the energy of a policy. Hence, the minimization of the H-term, $H(\theta)$, can reduce the traninig variance.
> >
> >     3) The authors have presented how the H-term guide the policy converge to high-quality local minima from multiple local minimas in the three examples of Fig. 1 (the case of $\gamma = 1$) and Fig. 6 (the case of $\gamma \in (0, 1)$).
> >
> >         The authors also showed that the H-term helps existing deep reinforcement learning algorithms converge to high-quality local minima. The conventional PPO algorithm (note that we also tested other DRL algorithms, DDPG, TD3, and SAC) randomly converges to one of several policies, resulting in high variance. This is empirically shown in the third column of Table 2. However, H-term as a regularization term guides those policies converge to a phyisically stable policy with low variances, as shown in the column 4-6 of Table 2.
> >
> >         [1] Shakir Mohamed, Mihaela Rosca, Michael Figurnov, and Andriy Mnih. Monte carlo gradient estimation in machine learning. J. Mach. Learn. Res., 21(132):1–62, 2020.
> >
> > - “Truncation of K”
> >
> >     The truncation of K is determined by the computation costs. The computation complexity of the proposed H-term,
> >     $$\hat{H}(\theta)= - \frac{1}{N'}\sum_{i = 1}^{N'} \left[\sum_{k = 0}^{K-1} \gamma^k\cdot R(\mu^i_k)\cdot  \nabla_\theta\log\left[\pi_\theta(\mu^i_0)\cdots\pi_\theta(\mu^i_k)\right]\right],$$ is $O(N'\frac{K(K+1)}{2})$. The computational cost is ignorable when a truncation of a small $K$ is employed, say $K=8, 16, 24$. The authors foresee a potentially high computational cost for future works if Bellman equations in RL would be replaced by the Hamiltonian equation. Note that the accuracy of the H-term approximation is directly related to the K-truncation. Therefore, future works may require a very accurate estimate, they may need a larger K, which may experience high computational costs.

---

> > > ### Author Response · Authors · 2022-11-17
> > > **Response to Reviewer 6uJh (3/3)**
> > >
> > >
> > > - “Notation of $H(\theta)$”
> > >     The authors agree the notation of $H$ in Section 4.3 is not clear. The proposed H-term is an energy of policy $H(\pi)$. We reparameterize $H(\pi)$ with $H(\theta)$. In section 4.3, the use of $H(0)$ and $H(1)$ is to differentiate the calculation of $\hat{H}(\theta)$ over trajectories started from state $0$, and those trajectories started from state $1$. We have modified this error in the updated version.
> > >
> > >     In Eqn (3), $H(θ) \triangleq  −E_{S_0,A_0}[Q_{\pi_\theta} (S_0, A_0)]$ is the definition of the Hamiltonian equation.
> > >
> > >     In Section 4.3, the example (a) and (b) have only two states. Hence the authors use $H(0)$ and $H(1)$ to differentiate the calculation of H-term on trajectories with initial state $0$ and the calculation of H-term on thrajectories with initial state $1$. In details, it is $$\sum_{k = 0}^{K-1} \gamma^k\cdot R(\mu_k)\cdot  \nabla_\theta\log\left[\pi_\theta(\mu_0)\cdots\pi_\theta(\mu_k)\right], \mu_0 = (\color{red}{s_0=0}, a_0),$$$$\sum_{k = 0}^{K-1} \gamma^k\cdot R(\mu_k)\cdot  \nabla_\theta\log\left[\pi_\theta(\mu_0)\cdots\pi_\theta(\mu_k)\right], \mu_0 = (\color{red}{s_0=1}, a_0).$$
> > >
> > > - “Computation complexity”
> > >     As mentioned above, the computation complexity refers to the inner computation of the Monte Carlo simulation of (5) and (6).
> > >
> > >      The compuation complexity of REINFORCE's policy gradient $$\frac{1}{N}\frac{1}{T}\sum_{i = 1}^{N}\left[\sum_{t=0}^{T-1} G_t^i\cdot  \nabla_\theta\log\pi_\theta(\mu^{i}_t)\right]$$ is $O(NT)$.
> > >
> > >      The computation complexity of Hamiltonian policy graident $$- \frac{1}{N'}\sum_{i = 1}^{N'} \left[\sum_{k = 0}^{K-1} \gamma^k\cdot R(\mu^i_k)\cdot  \nabla_\theta\log\left[\pi_\theta(\mu^i_0)\cdots\pi_\theta(\mu^i_k)\right]\right]$$ is $O(N'K(K+1)/2$.
> > >
> > >      By employing the H-term as a regularization term, the computation complexity becomes $$O(NT + N’K(K + 1) / 2).$$

---

### Decision · Program_Chairs · 2023-01-20

**Decision:**

Reject

**Justification For Why Not Higher Score:**

The main idea of the paper is not very convincing. A important connection to prior algorithm is not well discussed.

**Justification For Why Not Lower Score:**

N/A

**Metareview: Summary, Strengths And Weaknesses:**

The paper aims to address the instability issue of Deep Reinforcement Learning (DRL) and proposes a quantum K-spin Hamiltonian regularization term (called H-term) to help a policy network converge to a high-quality local minima. This new regularization term is inspired from the K-spin Ising model with energy minimization principle. Authors perform experiments by adding H-terms to both PPO and DDPG on MuJoCo tasks, which show that the proposed methods (1) are more likely to find better solutions, (2) have lower variance.

While the paper presents a new interesting idea, reviewers all have difficulty understanding the deep connections between deep reinforcement learning and K-spin Ising model. The theoretical arguments/derivations presented in the paper are not convincing. Multiple reviewers also pointed out the potential connection between the proposed methods with REINFORCE policy gradient, which is not clearly specified in the paper. We thus recommend rejection.